# Mitochondrial Stress Links Environmental Triggers with Pro-Inflammatory Signaling in Crohn’s Disease

**DOI:** 10.3390/antiox12122105

**Published:** 2023-12-13

**Authors:** Flores Martín-Reyes, Manuel Bernal, Cristina Rodríguez-Díaz, Damaris Rodríguez-de los Reyes, Ailec Ho-Plagaro, Francisca Rodríguez-Pacheco, Laura Camacho-Martel, Raquel Camargo-Camero, Francisco J. Rodríguez-González, Guillermo Alcain-Martínez, Rafael Martín-Masot, Víctor M. Navas-López, Marina Villanueva-Paz, María Isabel Lucena, Eduardo García-Fuentes, Carlos López-Gómez

**Affiliations:** 1Instituto de Investigación Biomédica de Málaga y Plataforma en Nanomedicina-IBIMA Plataforma BIONAND, 29590 Malaga, Spain; floresmr@uma.es (F.M.-R.); mbernal@uma.es (M.B.); cris.rdrz@gmail.com (C.R.-D.); damarisrodriguezdelosreyes@gmail.com (D.R.-d.l.R.); ailec_hp@hotmail.com (A.H.-P.); paqui.endocrino@gmail.com (F.R.-P.); laura.camachoma@gmail.com (L.C.-M.); raquelcamero@hotmail.com (R.C.-C.); galcainm@hotmail.com (G.A.-M.); rafammgr@gmail.com (R.M.-M.); victor.navas@gmail.com (V.M.N.-L.); marvp75@gmail.com (M.V.-P.); lucena@uma.es (M.I.L.); 2Unidad de Gestión Clínica de Aparato Digestivo, Hospital Universitario Virgen de la Victoria, 29010 Malaga, Spain; 3Departamento de Biología Molecular y Bioquímica, Facultad de Ciencias, Universidad de Málaga, 29010 Malaga, Spain; 4Sección de Gastroenterología y Nutrición Infantil, Unidad de Gestión Clínica de Pediatría, Hospital Regional Universitario de Málaga, 29010 Malaga, Spain; 5Servicio de Farmacología Clínica, Hospital Universitario Virgen de la Victoria, Departamento de Farmacología, Facultad de Medicina, Universidad de Málaga, 29010 Malaga, Spain; 6Centro de Investigación Biomédica en Red de Enfermedades Hepáticas y Digestivas, Instituto de Salud Carlos III, 28220 Madrid, Spain; 7UICEC IBIMA, Plataforma SCReN (Spanish Clinical Research Network), Servicio de Farmacología Clínica, Hospital Universitario Virgen de la Victoria, Universidad de Málaga, 29010 Malaga, Spain

**Keywords:** non-steroidal anti-inflammatory drugs, inflammatory bowel diseases, Crohn’s disease, mitochondria, mitochondrial unfolded protein response

## Abstract

Inflammatory Bowel Diseases (IBD) are a group of chronic, inflammatory disorders of the gut. The incidence and activity of IBD are determined by both genetic and environmental factors. Among these factors, polymorphisms in genes related to autophagy and the consumption of non-steroidal anti-inflammatory drugs (NSAIDs) have been consistently associated with IBD. We show that NSAIDs induce mitochondrial stress and mitophagy in intestinal epithelial cells. In an altered mitophagy context simulating that observed in IBD patients, NSAID-induced mitochondrial stress leads to the release of mitochondrial components, which act as Danger Associated Molecular Patterns with pro-inflammatory potential. Furthermore, colonic organoids from Crohn’s disease patients and healthy donors show activation of the mitochondrial Unfolded Protein Response (UPR^mt^) upon treatment with ibuprofen. Finally, colon biopsies from Crohn’s disease patients in remission or with low-to-moderate activity also show expression of genes involved in UPR^mt^, while patients with severe activity show no increase compared to healthy donors. Our results suggest the involvement of mitochondria in the mechanisms triggering inflammation in IBD after NSAID use. Moreover, our results highlight the clinical relevance of mitochondrial stress and activation of the UPR^mt^ pathway in the pathophysiology of Crohn’s disease.

## 1. Introduction

Inflammatory Bowel Diseases (IBD) are a group of chronic, relapsing disorders characterized by inflammation of the gastro-intestinal tract. IBD mainly encompasses Ulcerative Colitis (UC) and Crohn’s disease (CD). The incidence of IBD is increasing worldwide, and it is becoming one of the main causes of disability in the young population. It is currently accepted that IBD is triggered by environmental factors in genetically susceptible individuals. Genetic factors include but are not restricted to polymorphisms located in genes involved in autophagy, like ATG16L1, IRGM, or NOD2 [1]. Likewise, several environmental factors related to a “western life-style”, like diet, psychological stress, or perinatal antibiotic use, have been associated with IBD [2]. Among environmental factors, the use of non-steroidal anti-inflammatory drugs (NSAIDs) has been repeatedly associated with both a higher incidence of IBD [3] and higher activity of the disease [4]. Nevertheless, most studies are descriptive and do not investigate the mechanisms by which NSAIDs paradoxically activate inflammatory pathways leading to disease onset or relapses. Overall, progress has been made in unraveling both genetic and environmental factors, and functional studies based on previously reported genetic associations have shown the involvement of cellular mechanisms in IBD, as in the case of autophagy [1,5]. Unraveling mechanisms of gene-environment interaction leading to IBD should be our next step in achieving a full view of the molecular mechanism of the disease. However, these mechanisms remain poorly understood.

The involvement of mitochondria in IBD has been previously described: swollen (less efficient) mitochondria and low levels of adenosine triphosphate (ATP) have been associated with changes in ileum permeability [6]; reduction in oxidative phosphorylation is observed in colon from IBD patients [7]; and mitochondrial Unfolded Protein Response (UPR^mt^), a mitochondrial stress pathway, is activated in intestinal epithelial cells from patients with IBD [8]. In addition, it has been recently reported that mitochondrial DNA (mtDNA) is released during active IBD [9]. Mitochondrial components, such as mtDNA, cardiolipin, ATP, and n-formyl-peptides, can act as Danger Associated Molecular Patterns (DAMPs) and activate pro-inflammatory pathways [10]. Interestingly, mitochondrial dysfunction caused by environmental triggers associated with IBD, like gut microbiota [11], NSAIDs [12], or saturated fatty acids [13], are also linked to pro-inflammatory signaling. We then hypothesized that, under mitochondrial stress, impaired autophagy (and subsequently, impaired mitophagy) may lead to the release of mitochondrial DAMPs (mtDAMPs) with pro-inflammatory potential. In fact, it has been shown that mtDNA can activate the NLRP3 inflammasome [14], which is involved in IBD.

In the present study, we used in vitro models to assess the impact of NSAIDs on mitochondrial function and the release of mtDAMPs, as well as the levels of activation of UPR^mt^ in biopsies from CD patients and healthy donors.

## 2. Materials and Methods

### 2.1. Subjects

All subjects were recruited at the Gastroenterology Department of the Virgen de la Victoria University Hospital (Malaga, Spain). All participants gave their written informed consent; the study protocol was carried out in accordance with the ethical guidelines of the Declaration of Helsinki, and the study was approved by the Malaga Provincial Research Ethics Committee, Malaga, Spain (PI18/01652, PI0244-2021, ORG-Digest_2021). All subjects were older than 18 years. CD patients attending the Gastroenterology Department for a follow-up colonoscopy and healthy volunteers from the colon cancer screening program with non-pathological findings were recruited. Samples from subjects were obtained from normal-appearing mucosa and were processed immediately after their reception in the Virgen de la Victoria University Hospital Biobank (Andalusian Public Health System Biobank, Malaga, Spain). The demographic and clinical data of subjects are described in detail in Appendix A. Disease activity was assessed by the Harvey-Bradshaw index.

### 2.2. Cell Culture and Drugs

T84 cells (ATCC^®^ CCL-248™) were obtained from The European Collection of Authenticated Cell Cultures (ECACC). All reagents were from Sigma-Aldrich (Sigma-Aldrich, St. Louis, MO, USA) unless otherwise specified. T84 cells were cultured in DMEM:F12 15 mM HEPES media supplemented with 10% Fetal Bovine Serum (FBS), 1 mM L-glutamine, 100 IU/mL penicillin, and 100 μg/mL streptomycin. For experiments, T84 cells were maintained for 3 weeks after reaching confluency to induce differentiation towards functional enterocytes/colonocytes [15]. Cell media was changed every 2/3 days during the differentiation phase. The induction of differentiation in our cell model was assessed by gene expression of MS4A12 (Appendix A) [15]. After differentiation, cells were exposed to 1 mM of different drugs for 24 h: ibuprofen (I1892, Sigma-Aldrich), diclofenac (D6899, Sigma-Aldrich), naproxen (M1275, Sigma-Aldrich), celecoxib (PZ0008, Sigma-Aldrich), acetyl salicylic acid (PHR1003-1G, Sigma-Aldrich), paracetamol (6056776, Fresenius Kabi España, S.A.U., Barcelona, Spain) and paraquat (36541, Sigma-Aldrich). Noteworthy, 1 mM represents approximately 6× and 3.5 of ibuprofen [16] and naproxen [17] C_max_, respectively; therefore, the concentrations used in this study are biologically relevant. On the other hand, 1 mM represents approximately 134× of diclofenac C_max_ [18]. However, we considered keeping the same concentration for all drugs to better compare the effects. Furthermore, in vitro studies typically use up to 100× C_max_ to assess the effect of drugs [18,19]; therefore, this concentration would still remain experimentally relevant for the case of diclofenac.

One aliquot of colonic biopsies obtained during colonoscopy from patients with CD and healthy donors from the colorectal cancer screening program was immediately stored at –80 °C until analysis. Another aliquot of colonic biopsies was used to develop colonic organoids using Intesticult^TM^ Organoid Growth Medium (100-0190, Stemcell Technologies, Saint Égrève, France), following the manufacturer’s protocol.

### 2.3. Mitochondrial Function Assays

To measure mitochondrial membrane potential and Reactive Oxygen Species (ROS) production in T84 cells, T84 cells were seeded in a black bottom 96 well-plate at confluency, differentiated for three weeks, and stained with 800 nM tetramethylrhodamine methyl ester (TMRM) (TMRM Assay Kit, ab228569, Abcam, Cambridge, UK) for assessment of mitochondrial membrane potential or 5 µM dihydroethidine (DHE) (Assay Kit Reactive Oxygen Species, ab236206, Abcam, Cambridge, UK) for assessment of ROS production. FCCP (200 nM) and Antimycin A (10 µM) were used as controls in the TMRM and DHE assays, respectively. Fluorescence was read in an FL600 fluorescent reader (BioTek Instruments, Inc., Winooski, VT, USA). Within each assay, different conditions were run in triplicates. A total of 5 experiments were analyzed for each assay.

### 2.4. Gene Expression

RNA was isolated using QIAzol (QIAGEN Science, Hilden, Germany) following manufacture’s procedure and quantified by spectrophotometry (Nanodrop 2000, Thermo Fisher Scientific, Waltham, MA, USA). cDNA was synthesized using M-MLV reverse transcriptase (Promega, Madison, WI, USA) from 500 ng of RNA. Gene expression was assessed by qPCR in a LightCycler 480 (Roche Diagnostics S.L, Barcelona, Spain) using 1 µL from a 1/5 dilution of the cDNA, SensiFAST™ SYBR^®^ Hi-ROX (Bioline, London, UK) and the forward (F) and reverse (R) primers detailed in Appendix A. The ddCt method was used to analyze the data. S18 was used as a housekeeping gene, and data were referenced to % of the expression in the untreated samples. Within each assay, each sample was run in triplicates (technical triplicates), and triplicate averages were used for statistical analysis. Within each assay, 3 samples/conditions were run (biological replicates), and a total of 3 assays (total of 9 values/conditions) were analyzed.

### 2.5. Allelic Discrimination Assay

Allelic discrimination assays to assess the ratio of cells carrying the reference and risk alleles for the T300A polymorphism (rs2241880) in ATG16L1 gene were performed by the Genomic core of IBIMA. Samples of T84 cells used for other experiments throughout the study were assessed, and DNA was isolated using the salting-out method. Afterward, samples were sequenced in a PyroMark Q96 ID (QIAGEN Science, Hilden, Germany) using the following primers (sequence 5′–3′): forward: CCCCCAGGACAATGTGGA; reverse: CGAAGACACACAAGGCAGTAGCT; sequencing primer: TTTACCAGAACCAGGAT. The reaction was run using the following conditions: 95 °C for 15 min, followed by 45 cycles of 94 °C for 30 s, 56 °C for 30 s, and 72 °C for 2 s, and a final step of 72 °C for 10 min.

### 2.6. Immunoblotting

Cells were detached using Trypsin-EDTA and spun at 300× *g* for 10 min. Cell pellets were resuspended in lysis buffer (0.5 M Tris-HCl, pH 6.8; 12% SDS; 10% glycerol; 5% β-mercaptoethanol; 0.2% bromo-phenol blue). Samples were denaturalized at 95 °C for 5 min. Protein samples were measured with the NanoDrop^TM^ One (Thermo Fisher Scientific, Waltham, MA, USA) using the Abs280 preprogrammed direct absorbance, and 30 μg of total protein were loaded into 4–15% gradient Mini-PROTEAN^®^ TGX^TM^ Stain-free gels (Bio-Rad Laboratories, Hercules, CA, USA). Gels were transferred to nitrocellulose membrane, 1 h at 4 °C 100 V constant, and blocked 1 h at room temperature in TBS-T buffer (Tris 20 mM, NaCl 137 mM, Tween-20 0.1%) containing 5% non-fatty dry milk. Membranes were hybridized with rabbit anti-LC3B 1:1000 (AB192890, Abcam, Cambridge, UK) overnight at 4 °C. Membranes were washed in TBS-T and incubated 1 h at room temperature with goat anti-rabbit IgG (AB97051, Abcam, Cambridge, UK) at 1:10,000 dilution. Immunoreactive bands were detected with SuperSignal^TM^ West Pico Chemiluminescent Substrate (Thermo Fisher Scientific, Waltham, MA, USA) in a Chemidoc XRS System (Bio-Rad Laboratories, Hercules, CA, USA). Bands were quantified and analyzed by Fiji software (ImageJ2, version 2.14.0) [20]. Values were normalized using 4 different non-specific bands from the stain-free image.

### 2.7. Assessment of mtDAMPs

After treatment with different drugs, cell media and T84 cells were collected. Cytoplasmic fractions were isolated from cells using differential centrifugation. In short, cell media from differentiated T84 cells growing in a 6-well plate without treatment or under ibuprofen (1 mM) or paraquat (1 mM) treatment was collected, and cells were scraped and collected in MTSE buffer (210 mM d-mannitol, 70 mM sucrose, 10 mM Tris HCl pH 7.5, 0.2 mM EGTA). A total of 100 µL from the homogenate was stored at −20 °C (whole fraction), and the remaining was centrifuged at 1000× *g* for 5 min at 4 °C. Pellet (nuclei fraction) was stored at −20 °C and supernatant was collected and centrifuged at 13,000× *g* for 2 min at 4 °C. Both supernatant (cytosolic fraction) and pellet (mitochondrial fraction) were separated and stored at −20 °C. Presence of mtDNA in the cell media, cytoplasmic fraction, and whole fraction were assessed by qPCR using the AB7500 (Thermo Fisher Scientific, Waltham, MA, USA), the Universal TaqMan^®^ qPCR master Mix (4304437, Thermo Fisher Scientific, Waltham, MA, USA), and TaqMan^®^ assays targeting mtDNA (mtRNR1, 4331182-FAM, Thermo Fisher Scientific, Waltham, MA, USA) and nuclear DNA (nDNA) (RNase P, 4316844-VIC, Thermo Fisher Scientific, Waltham, MA, USA). Levels of mtDNA in both cell media and cytosolic fractions were normalized by nDNA levels in whole fractions to correct for differences in the cell number contained in the samples. In addition, to correct for possible differences in the amount of mtDNA in different cell samples, which could also bias the results, we performed a second analysis normalizing mtDNA levels in both cell media and cytosolic fraction to mtDNA levels per cell (mtDNA/nDNA) in whole fraction.

Cardiolipin was measured in cytosolic fraction using a commercial kit (Cardiolipin Assay Kit, ab241036, Abcam, Cambridge, UK) following the manufacturer protocol. Levels of cardiolipin were expressed as nmol and normalized to mg of protein concentration in each sample (determined using a Bradford assay).

### 2.8. Statistical Analysis

Data grouped in columns are expressed as the mean ± standard deviation (SD) when all groups are normally distributed or as the median ± interquartile range (IQR) when several groups deviate from a normal distribution. The Kolmogorov–Smirnov test was used to assess the normal distribution of the data. A minimum of three experiments per group was used. For data grouped in columns and normally distributed, ANOVA, Dunnett’s test for multiple comparisons, or Mann–Whitney test were used. For data grouped in columns and not normally distributed, Kruskal–Wallis test and Dunn’s test for multiple comparison or Mann–Whitney test was used. For paired data (gene expression within each organoid line), a Wilcoxon text was used. To detect differences between organoid lines from healthy controls (HC) or CD patients, a two-way ANOVA was performed. Spearman’s correlation test was used to assess the relationship between different variables. A *p* value of <0.05 was statistically significant.

## 3. Results

### 3.1. NSAIDs Induce Mitochondrial Stress in an In Vitro Model of Intestinal Epithelium

To assess whether NSAIDs induce mitochondrial stress, we treated T84 cells with non-selective cyclo-oxygenase (COX) inhibitors (ibuprofen and diclofenac), selective COX-2 inhibitor (celecoxib), and a non-COX inhibitor analgesic (paracetamol). A dramatic increase in the mitochondrial membrane potential was observed in cells treated with either ibuprofen or diclofenac, compared to untreated cells, which suggests a higher demand for ATP in conventional NSAIDs-treated cells (Figure 1A). In contrast, cells treated with celecoxib or acetyl salicylic acid showed no significant differences, and cells treated with paracetamol only showed a slightly reduced mitochondrial membrane potential.

A higher production of ROS was also observed in cells treated with diclofenac, likely a consequence of a higher activity of the mitochondrial electron transport chain (Figure 1B). In accordance with data from membrane potential, cells treated with celecoxib or acetyl salicylic acid showed no significant difference compared to untreated cells, and cells treated with paracetamol showed a mild increase in ROS production. Thus, conventional NSAIDs induce a higher demand for ATP, subsequently increasing the production of ROS, which altogether represents mitochondrial stress.

To confirm this, we assessed the activation of UPR^mt^, a mitochondrial stress response. For that, we focused on NSAIDs, which showed an effect on mitochondrial membrane potential, ibuprofen, and diclofenac. In addition, we assessed the effects of naproxen, which, as ibuprofen, is a derivative of propionic acid, and we used paraquat (a complex I inhibitor) as a positive control of the activation of UPR^mt^. Cells treated with conventional NSAIDs showed significantly higher levels of the UPR^mt^ transcripts (Figure 2). More specifically, ibuprofen increased levels of the UPR^mt^ transcripts ATF4 and CHOP, diclofenac increased levels of CHOP, and naproxen increased the levels of ATF4. Subsequently, NSAIDs increased the levels of the mitokine GDF15 (ibuprofen and naproxen) as well as the levels of PGC1α, a marker of mitochondrial biogenesis (ibuprofen and naproxen). Gene expression levels of the mitochondrial protease LON were also altered by treatment with ibuprofen (Appendix A). The gene expression of TRAP1 was also altered by different treatments, as denoted by a significant result from a Kruskal–Wallis test (Appendix A). Although no significant result was observed in Dunn’s test compared with untreated cells, TRAP1 shows a trend towards a decrease upon treatment with NSAIDs.

### 3.2. Under Impaired Mitophagy, Mitochondrial Stress Leads to Release of mtDAMPs

The T300A polymorphism (rs2241880) in ATG16L1 is one of the most strongly associated genetic variations with IBD [1,21]. It has been previously shown that this polymorphism impairs autophagy [21,22], likely by increasing the sensitization of the T300A allele to Caspase-3 degradation [5]. Because the T84 is a cancer cell line, it contains a number of chromosomic alterations, one of which is the lack of one copy of chromosome 2, in which ATG16L1 is located. Therefore, T84 cells are homozygous for the rs2241880 alleles in ATG16L1. We found that our sample of T84 cells contained a mixed population of cells carrying different alleles. Using pyrosequencing, we analyzed 11 different samples of T84 cells and observed that 69.41% of the T84 cells are homozygous for the risk allele (C) and 30.59% are homozygous for the T allele (Table 1).

Treatment with ibuprofen increased both LC3B-I and LC3B-II isoforms in T84 cells (Figure 3). In fact, treatment with ibuprofen increased the conversion of LC3B-I to LC3B-II, which is a hallmark of autophagy. In contrast, treatment with naproxen did not significantly increase the levels of LC3B-I and LC3B-II.

Since only ibuprofen showed significant activation of autophagy, we next evaluated the effects of treatment with ibuprofen on the release of mtDAMPs in this context of altered autophagy, in which most of the cells are homozygous for the T300A variation. We observed that mtDNA levels increased in the extracellular media when we normalized to cell number (Figure 4A). When we normalized to levels of mtDNA/cell in the whole fraction, both mtDNA levels in cytosolic fractions and the extracellular media were increased (Figure 4B). Consistently, levels of cardiolipin were significantly increased in the cytosolic fractions of T84 cells (Figure 4C).

### 3.3. Ibuprofen Activates UPR^mt^ in Colonic Organoids

To evaluate the relevance of this mechanism in CD, we aimed to validate our results in a non-carcinogenic in vitro model. For this purpose, colonic organoids were developed from colon biopsies obtained from patients with CD and healthy donors. Organoids from both CD patients and healthy controls treated with ibuprofen showed an increase in the UPR^mt^ transcripts ATF4 and ATF5, as well as the mitokine GDF15, compared to untreated organoids (Figure 5). Levels of TRAP1 were mildly decreased upon treatment with ibuprofen, which would resemble the results obtained with the T84 cells (Appendix A). We did not observe significant differences between organoids developed from patients and healthy controls in any treatment group (Appendix A).

To evaluate the relationship between different markers of UPR^mt^, we ran a correlation analysis of gene expression under the different experimental conditions previously assessed (Table 2). The three UPR^mt^ transcripts (ATF4, ATF5, and CHOP) strongly correlated with each other, as well as FGF21 and PGC1α. However, the expression of GDF15 correlated only with ATF4 and ATF5 and not with CHOP. The chaperone TRAP1 showed moderate correlation with the mitokines GDF15 and FGF21 and strongly correlated with the mitochondrial protease LON. The latter also showed a correlation with UPR^mt^ transcripts (ATF4 and ATF5), mitokines (GDF15 and FGF21), and PGC1α.

### 3.4. UPR^mt^ Is Activated in the Colon of Patients in Remission and Low-to-Moderate Activity

The gene expression of transcripts involved in UPR^mt^ was assessed in colon biopsies obtained from patients with CD and healthy donors. Compared to healthy controls and patients with severe activity, patients in remission or with low or moderate activity showed increased gene expression of ATF4, ATF5, and CHOP (Figure 6). Significant changes were observed between patients in remission and patients with severe activity for ATF4, between healthy controls and patients in remission for ATF5, and between patients with low or moderate activity and patients with severe activity for CHOP.

In addition, a trend towards increased expression of DRP1, a gene involved in mitochondrial fission, and STAT1, a gene involved in inflammation, were also observed in patients in remission and with low-to-moderate activity (*p* = 0.0517 and *p* = 0.0556 in Kruskal–Wallis test, respectively) (Appendix A). As expected, correlation analysis shows a strong correlation between the three UPR^mt^ transcripts (Appendix A) but also moderate correlations between UPR^mt^ transcripts (ATF4, ATF5, and CHOP) and expression of genes involved in mitochondrial fission (FIS1 and OPA1). In fact, ATF5 and FIS1 are strongly correlated, which suggests a potential role for UPR^mt^ in inducing early mitophagy mechanisms. In addition, we found a strong positive correlation between the expression of TRAP1 and two genes involved in the Interferon (IFN) signaling pathway, STAT1 and OAS1, and a strong positive correlation between DRP1 and NF-κB.

## 4. Discussion

Environmental risk factors play an important role in both the onset of CD and the triggering of disease exacerbation in patients with CD. A large volume of literature has associated different environmental factors with CD [2]. However, most of the studies are descriptive, and thus, the mechanisms linking the exposition to environmental factors and disease symptoms remain largely unknown. Mitochondrial dysfunction in CD has been long known, but studies fail to describe the cellular mechanisms explaining the association between the alteration of mitochondrial homeostasis and disease activity. In the present study, we provide evidence of a mechanism involving mitochondria in the signal transduction from a bona fide risk factor for CD, treatment with NSAIDs, and activation of pro-inflammatory pathways.

As expected, NSAIDs induced mitochondrial stress in the T84 cell line, which is denoted by an increased mitochondrial membrane potential and subsequently increased ROS production. However, to our knowledge, this is the first report describing activation of the UPR^mt^, a mitochondrial stress response, after treatment with NSAIDs. Interestingly, only non-selective COX inhibitors (conventional NSAIDs) induced mitochondrial stress, while a selective COX-2 inhibitor (celecoxib) and acetyl salicylic acid, which are not associated with a higher incidence of CD [3], did not induce mitochondrial stress. It is worth noting that celecoxib is considered to be safe for patients with CD, as opposed to conventional NSAIDs. NSAIDs-treated cells also showed increased levels of the mitokine GDF15, as well as PGC1α, a marker of mitochondrial biogenesis. These proteins are downstream players of the UPR^mt^, and their elevation confirms the activation of mitochondrial response-to-stress pathways. Particularly, GDF15 has garnered much attention in recent years. As a mitokine, levels of GDF15 are increased upon mitochondrial stress [23]. However, increased levels of GDF15 are found not only in primary mitochondrial disorders [24] but also in different diseases like neurodegenerative diseases [25], cardiovascular diseases [26], and cancer [27]. Furthermore, GDF15 is elevated in conditions like aging [28] or exercise [29]. GDF15 has been previously related to CD. A higher serum GDF15 level in patients with CD was significantly associated with a low skeletal muscle index, which is related to the intestinal complications of CD. The complex role of GDF15 during physiological and pathophysiological conditions is still a matter of debate as the beneficial and detrimental effects of GDF15 are described [23].

To further investigate the cellular mechanisms linking environmental CD-risk factors with genetic predisposition to CD, we observed that our in vitro cell model resembled part of the genetic background of CD patients, which is altered autophagic mechanisms. More specifically, a high percentage of our T84 cells are homozygous for the CD-risk variant T300A in the ATG16L1 gene, which has been consistently associated with CD [21]. Under conditions of mitochondrial stress, mitophagy (autophagy of mitochondria) is activated, with the aim of removing dysfunctional mitochondria and recycling its components. A hallmark of autophagy is the conjugation of LC3B-I to phosphatidylethanolamine on the surface of autophagosomes, generating the lipidated LC3B-II form. This process is driven by ATG16L1, which localizes on the source membranes for autophagosome formation. In fact, ATG16L1 knock-out mice are incapable of converting LC3B-I to LC3B-II [22,30]. Interestingly, the CD-associated SNP rs2241880 (ATG16L1, T300A) is only partially functional, and mice carrying this polymorphism show reduced levels of LC3B-II [22], likely because of a higher sensitization of the T300A allele to be degraded by caspase 3 [5], leading to reduced levels of ATG16L1 available to convert LC3B-I to LC3B-II.

Likely as a consequence of the mitochondrial stress, treatment with ibuprofen activated autophagy in the T84 cell line, as observed by the increased levels in LC3B-II. This observation is remarkable given the fact that only 30% of our T84 cells carry a wild-type allele of ATG16L1 and contribute to the formation of LC3B-II, while 70% of cells carry the CD-risk variant T300A, which impairs the conversion of LC3B-I to LC3B-II. Notably, the treatment of T84 cells with NSAIDs led to the release of mtDAMPs, as observed by increased mtDNA levels in both the cytosol and the extracellular media and increased cardiolipin levels in the cytosol. These data suggest a collapse of the mechanism of autophagosome formation after treatment with NSAIDs, specifically in T84 cells carrying the ATG16L1 T300A allele, due to the impaired lipidation of LC3B-I. As a consequence, improperly processed, dysfunctional mitochondria release their content to both the cytosol and the extracellular media.

Our results confirm previous studies observing activation of UPR^mt^ in the mucosa of patients with IBD [8] and provide a mechanistic rationale for the previous observation of mtDNA levels in blood with disease activity [9], which is the novelty of our study. Thus, our findings place mitochondrial stress as a link between genetic and environmental risk factors for CD. To confirm this mechanism in a non-carcinogenic cell model, we studied the activation of the UPR^mt^ in colonic organoids derived from CD patients. The elevation of ATF4, ATF5, and GDF15 upon treatment with ibuprofen confirms the effect of NSAIDs on mitochondrial homeostasis. Furthermore, although our study was not designed to assess the role of each transcription factor on target genes, our correlation analysis may suggest that the expression of GDF15 is not CHOP-dependant but rather ATF4- and ATF5-dependant. In fact, a previous study found that CHOP is dispensable for exercise-induced expression of GDF15 [29]. Furthermore, levels of TRAP1 decrease after treatment with ibuprofen. TRAP1 is a mitochondrial matrix chaperone reported to play a role in the activation of the UPR^mt^. It has been described that the modulation of the expression of TRAP1 activates the UPR^mt^ [31]; therefore, the downregulation in TRAP1, as observed upon treatment with NSAIDs, may be the trigger of the UPR^mt^. We did not find a strong negative correlation between TRAP1 and any of the UPR^mt^ transcripts, although we found a moderate positive correlation between TRAP1 and the mitokines GDF15 and FGF21. Therefore, our data support the role of the downregulation of TRAP1 in the response to mitochondrial stress upon treatment with NSAIDs. Interestingly, the upregulation of TRAP1 has been associated with therapeutic response to infliximab in CD [32], but the exact mechanism by which TRAP1 induces mucosal healing was not investigated. Thus, further studies assessing the role of TRAP1 and UPR^mt^ in mucosal healing are needed.

To assess the presence of mitochondrial stress in vivo, we studied the gene expression profile in colon biopsies from healthy donors and CD patients with different degrees of disease activity. Interestingly, we observed that UPR^mt^ transcripts in colon biopsies obtained from CD patients in remission or with low-to-moderate disease activity are increased compared not only to healthy controls but also to CD patients with severe activity. The role of UPR^mt^ in mitohormesis may be a plausible explanation for this observation. Hormeses (or mitohormesis for mitochondrial hormesis) are adaptive mechanisms aimed at counteracting the effects of repeated or chronic mild stress. The absence of activation of the UPR^mt^ may contribute to a higher activity index in certain CD patients. However, the absence of UPR^mt^ may also be the consequence of a more hostile cytoenvironment where higher levels of inflammation and the subsequent oxidative stress collapse cytoprotective pathways like the UPR^mt^. Supporting this, a recent study observed that IL-1β induced the maximum activation of UPR^mt^ in primary mouse chondrocytes at 10 ng/mL at 24 h. However, when the dose or time of exposure was increased, expression levels of UPR^mt^ transcripts decreased [33].

The level of DRP1, a gene involved in mitochondrial fission, was also elevated in patients in remission or with low or moderate activity. Mitochondrial fission can be considered the first step in the process of mitophagy [34]. In fact, UPR^mt^ transcripts correlated with the expression of genes involved in mitochondrial fission, confirming the described link between UPR^mt^, mitophagy, and mitochondrial fission [35]. Furthermore, the correlation of genes involved in inflammation (NF-ΚB, STAT1, OAS1) with genes involved in mitochondria quality control (TRAP1) or mitochondrial fission (DRP1) may link mitochondrial homeostasis and inflammatory signals.

## 5. Conclusions

Overall, our study describes, for the first time, a cellular mechanism linking an environmental risk factor (NSAIDs), pro-inflammatory signals (mtDAMPs), and mitochondrial stress in the context of CD. Furthermore, our study provides a mechanistic rationale for previous studies observing higher mtDNA levels in plasma from IBD patients [9] and the activation of UPR^mt^ in the mucosa of IBD patients [8]. Mitochondrial stress and the activation of the UPR^mt^ may also be behind the association of other environmental risk factors for IBD. Thus, further studies are needed to better understand the exact role of UPR^mt^ in IBD.

## Figures and Tables

**Figure 1 antioxidants-12-02105-f001:**
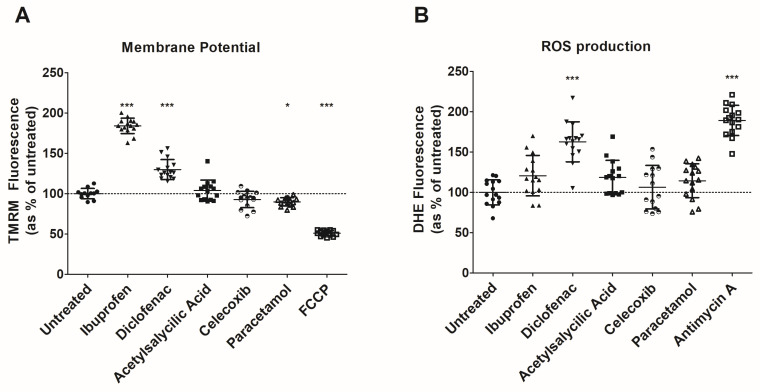
Mitochondrial membrane potential and Reactive Oxygen Species (ROS) production in T84 cells. (**A**) Data of mitochondrial membrane potential. Fluorescence (arbitrary units) was normalized to percentage of untreated cells. FCCP was used as negative control in the assessment of mitochondrial membrane potential since it is known to uncouple mitochondrial oxidative phosphorylation. Mean ± standard deviation is represented. The result in the ANOVA was *p* < 1.0 × 10^−4^. * indicates *p* < 0.05 and *** indicates *p* < 1.0 × 10^−3^ in a Dunnett’s test for multiple comparisons. (**B**) Data of ROS production. Fluorescence (arbitrary units) was normalized to percentage of untreated cells. Antimycin A (complex III inhibitor) was used as a positive control in the assessment of ROS production. Mean ± standard deviation is represented. The result in the ANOVA was *p* < 1.0 × 10^−4^. *** indicates *p* < 1.0 × 10^−3^ in a Dunnett’s test for multiple comparisons.

**Figure 2 antioxidants-12-02105-f002:**
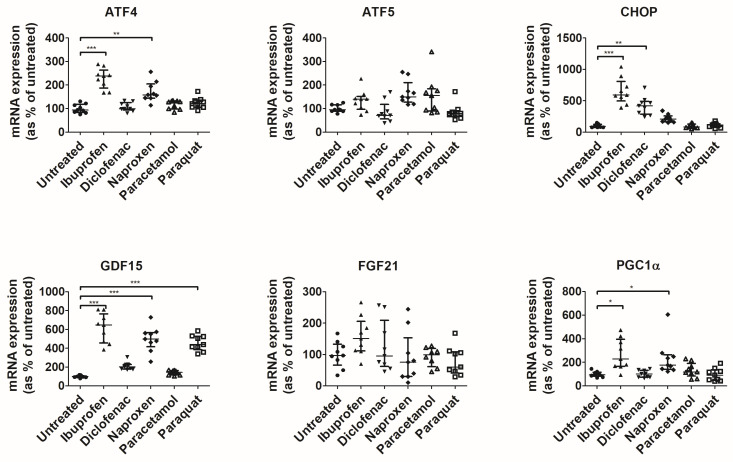
Expression of genes involved in mitochondrial response to stress in T84 cells. ddCt values were normalized to percentage of untreated cells. Median ± IQR is represented. Significant results from Kruskal–Wallis tests were ATF4, *p* < 1.0 × 10^−4^; ATF5, *p* = 6.0 × 10^−4^; CHOP, *p* < 1.0 × 10^−4^; GDF15, *p* < 1.0 × 10^−4^; and PGC1α, *p* = 2 × 10^−4^. * indicates *p* < 0.05, ** indicates *p* < 0.01 and *** indicates *p* < 1.0 × 10^−3^ in a Dunn’s test for multiple comparisons.

**Figure 3 antioxidants-12-02105-f003:**
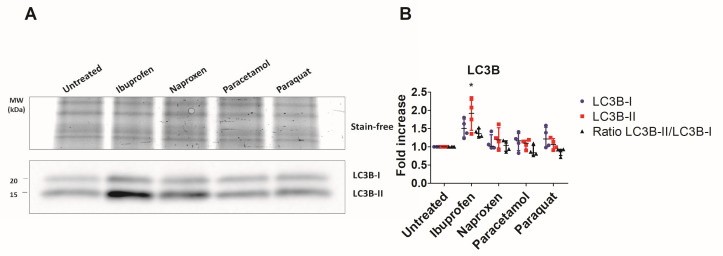
Assessment of different LC3B isoforms by immunoblot. (**A**) An image of a representative immunoblot membrane is shown. The lower block represents bands of LC3B-I and LC3B-II, and the upper block represents the stain-free image of unspecific proteins used to normalize. (**B**) Values from 4 different immunoblots. Each bar represents median ± IQR. Values were calculated as indicated in Methods. Significant results from Kruskal–Wallis test were LC3B-II (*p* = 0.0371); Ratio LC3B-II/LC3B-I (*p* = 0.0127). * indicates significant results in Dunn’s multiple comparison test compared to untreated cells.

**Figure 4 antioxidants-12-02105-f004:**
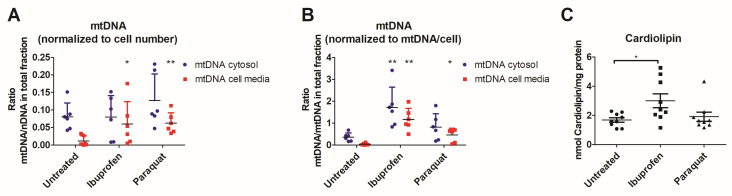
mtDAMPs levels in cytosol and cell culture supernatant. (**A**,**B**) represents mitochondrial DNA levels were calculated as a ratio between ddCt of a mitochondrial gene (mtRNR1) and ddCt of a nuclear gene (RNase P). (**A**) represents values normalized to ddCt of nuclear DNA in the whole fraction. Results from Mann–Whitney tests compared to untreated cells were extracellular media (ibuprofen, *p* = 0.04; paraquat, *p* = 2.2 × 10^−3^). (**B**) represents values normalized to mitochondrial DNA levels (ratio of mitochondrial and nuclear DNA) in the whole fraction. Results from Mann–Whitney tests compared to untreated cells were extracellular media (ibuprofen, *p* = 2.2 × 10^−3^; paraquat, *p* = 0.015); cytosolic fraction (ibuprofen, *p* = 2.2 × 10^−3^). (**C**) represents levels of cardiolipin in cytosolic fractions). Cardiolipin levels were normalized to mg of protein in each sample. Significant result from a Mann–Whitney test compared to untreated cells was ibuprofen, *p* = 0.0244. For the three panels, * indicates *p* < 0.05 and ** indicates *p* < 0.01.

**Figure 5 antioxidants-12-02105-f005:**
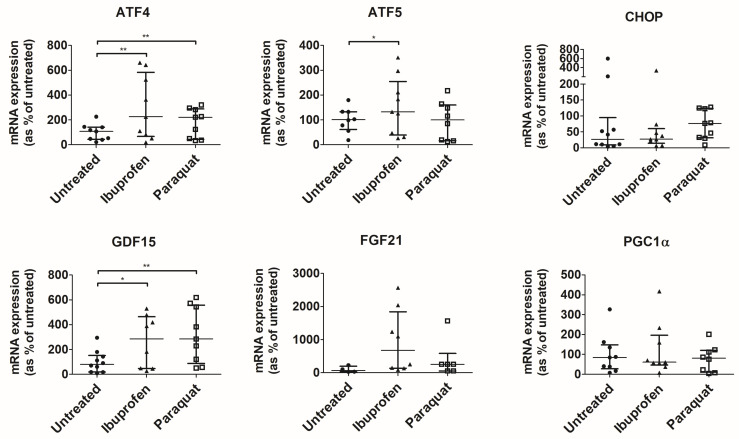
Expression of genes involved in mitochondrial response to stress in colonic organoids. ddCt values were normalized to percentage of untreated cells. Median ± IQR is represented. Data from organoids derived from patients and healthy controls are combined. Results from significant Wilcoxon tests comparing to untreated organoids were ATF4 (ibuprofen, *p* = 3.9 × 10^−3^; paraquat, *p* = 7.8 × 10^−3^); ATF5 (ibuprofen, *p* = 0.0469); GDF15 (ibuprofen, *p* = 0.0195; paraquat, *p* = 3.9 × 10^−3^). * indicates *p* < 0.05 and ** indicates *p* < 0.01 in Wilcoxon test.

**Figure 6 antioxidants-12-02105-f006:**
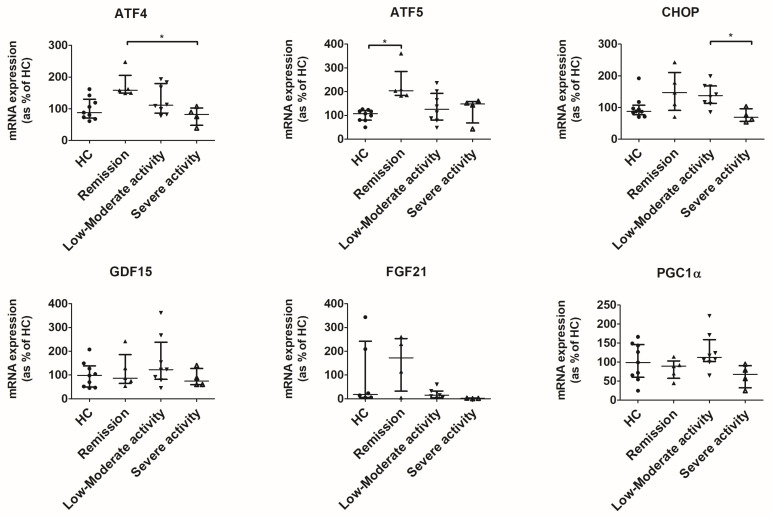
Expression of genes involved in mitochondrial response to stress in colonic biopsies. ddCt values were normalized to percentage of healthy donors. Median ± IQR is represented. Significant results from Kruskal–Wallis tests were ATF4, *p* = 0.0191; ATF5, *p* = 0.0144; and CHOP, *p* = 0.0234. * indicates *p* < 0.05 in a Dunn’s test for multiple comparisons.

**Table 1 antioxidants-12-02105-t001:** Allelic determination in rs2241880.

Replicate	% of T	% of C
1	29.64	70.36
2	35.43	64.57
3	30.35	69.65
4	32.16	67.84
5	29.22	70.78
6	31.59	68.41
7	29.85	70.15
8	30.77	69.23
9	26.00	74.00
10	31.18	68.82
11	30.27	69.73
Mean ± SD	30.59 ± 2.28	69.41 ± 2.28

Allelic discrimination was performed across 11 different samples of T84 cells used in other assays throughout the study. Allelic discrimination was performed on the complementary strand (T > C) to assess the ratio of cells carrying the non-risk allele (A) and the risk allele (G) in the coding strand (switch from Threonine to Alanine in the protein sequence).

**Table 2 antioxidants-12-02105-t002:** Correlation between expression of different UPR^mt^ transcripts in colonic organoids.

		ATF4	ATF5	CHOP	GDF15	FGF21	PGC1α	TRAP1	LON
ATF4	Pearson’s r		0.745	0.539	0.71	0.798	0.63	0.208	0.73
	*p* value		<0.001	0.004	<0.001	<0.001	0.001	0.297	<0.001
ATF5	Pearson’s r	0.745		0.636	0.613	0.811	0.779	0.285	0.579
	*p* value	<0.001		0.001	0.001	<0.001	<0.001	0.167	0.002
CHOP	Pearson’s r	0.539	0.636		0.243	0.613	0.862	0.015	0.342
	*p* value	0.004	0.001		0.222	0.007	<0.001	0.94	0.087
GDF15	Pearson’s r	0.71	0.613	0.243		0.702	0.49	0.462	0.834
	*p* value	<0.001	0.001	0.222		0.001	0.013	0.017	<0.001
FGF21	Pearson’s r	0.798	0.811	0.613	0.702		0.809	0.472	0.622
	*p* value	<0.001	<0.001	0.007	0.001		<0.001	0.048	0.006
PGC1α	Pearson’s r	0.63	0.779	0.862	0.49	0.809		0.127	0.513
	*p* value	0.001	<0.001	<0.001	0.013	<0.001		0.536	0.009
TRAP1	Pearson’s r	0.208	0.285	0.015	0.462	0.472	0.127		0.606
	*p* value	0.297	0.167	0.94	0.017	0.048	0.536		0.001
LON	Pearson’s r	0.73	0.579	0.342	0.834	0.622	0.513	0.606	
	*p* value	<0.001	0.002	0.087	<0.001	0.006	0.009	0.001	

Correlation between expression of assessed genes in colonic organoids. For each pair of genes, Pearson’s R and *p* value are shown. Color scale from red to green represents degree of correlation (red indicates Pearson’s r closer to 1, and green indicates Pearson’s r closer to 0).

## Data Availability

The data presented in this study are available from the corresponding author upon reasonable request.

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
