# Peer review of "Mitochondrial Stress Links Environmental Triggers with Pro-Inflammatory Signaling in Crohn’s Disease"

_antioxidants, 2023, doi:10.3390/antiox12122105_

Round 1
Reviewer 1 Report
Comments and Suggestions for Authors
The manuscript of Flores Martín-Reyes et al. reports that genes related to autophagy and consumption of non-steroidal anti-inflammatory drugs (NSAIDs) associate at IBD induce mitochondrial stress and mitophagy in intestinal epithelial cells. In particular, NSAID-induced mitochondrial stress leads to the release of mitochondrial components which act as Danger Associated Molecular Patterns with pro-inflammatory potential. Several experimental data reported by authors suggest the involvement of mitochondria in the mechanisms triggering inflammation in IBD after NSAIDs use.
In my opinion, the manuscript can be considered for publication. However, to confirm the mitochondrial stress induced by the NSAIDs I require at least a colorimetric assay (various commercial kits are available) in addition to the measurement of the mitochondrial membrane potential.
Author Response
The manuscript of Flores Martín-Reyes et al. reports that genes related to autophagy and consumption of non-steroidal anti-inflammatory drugs (NSAIDs) associate at IBD induce mitochondrial stress and mitophagy in intestinal epithelial cells. In particular, NSAID-induced mitochondrial stress leads to the release of mitochondrial components which act as Danger Associated Molecular Patterns with pro-inflammatory potential. Several experimental data reported by authors suggest the involvement of mitochondria in the mechanisms triggering inflammation in IBD after NSAIDs use.
In my opinion, the manuscript can be considered for publication. However, to confirm the mitochondrial stress induced by the NSAIDs I require at least a colorimetric assay (various commercial kits are available) in addition to the measurement of the mitochondrial membrane potential.
Response: The authors would like to thank the reviewer for its consideration of this article for publication. The reviewer suggested to perform one more assay to confirm NSAID-induced mitochondrial stress. However, we consider it may not be necessary. NSAIDs are already described to induce the production of ROS through mitochondrial stress [PMID 27049794, PMID 34499283 and 19049974 are just three examples]. In our cell model, alteration of mitochondrial membrane potential is perceived as a significant increase, likely indicating a higher demand of ATP. Elevated production of ROS is a consequence of an elevated mitochondrial activity. Our observation of higher levels of ROS in response to NSAIDs is consistent with the literature. In addition, to confirm mitochondrial stress, we measured mRNA levels of transcripts involved in the mitochondrial Unfolded Protein Response, which is a response to stress pathway. We consider the generalized increased in these transcripts (ATF4, CHOP and GDF15) consistently confirm the existence of mitochondrial stress in cells treated with conventional NSAIDs.

Reviewer 2 Report
Comments and Suggestions for Authors
Major comments
· Did the authors measure classic inflammation markers in response to the NSAID treatments? E.g. IL-6, IL-8, IL1b, TNF-a, COX-2, PGES, IL-10, CCL-20,… This seems to be missing as control for the actual effect of the treatments on the organoids. Can the authors speculate about the pathophysiology of the inflammation incurred in the genesis of IBD (autoimmune) and the effect of the NSAIDs on inflammation via mitochondrial stress?
· Following up on this, it would be very interesting to have included in the panel of drugs also Mesalazin, which also has NSAID activity, but is used as actual treatment for IBD.
· Why are Figures 3B as well as 4A and 4B shown as histograms instead of as individual datapoints? Please show all individual datapoints in all graphs (like in the other figures).
· For non-normal-distributed data (the authors mention that they use a normality test in the methods), median+-IQR should be shown instead of mean+-SD (the SD only means something if the data are normal distributed).
· The authors should use the correct statistical tests for multiple comparisons. Specifically, the authors use multiple t-tests or Mann-Whitney U tests to compare their different treatments with the same control group. However, this does not should never be done, as it increases the likelihood of committing a type 1 error. The correct tests to use is an ANOVA with an appropriate post-hoc test (e.g. Dunnett) for normal distributed data and Kruskal-Wallis with Dunn’s for non-normal distributed data.
· In Figure 3B, the authors state they use a paired-T-test. The same issue with multiple comparisons as described above applies here, but this graph presents yet an additional problem. The Untreated control is shown as being normalized to 1. If a group does not have a standard deviation, (i.e. all values = 1 in this group), they cannot be analyzed using a parametric test. The data should instead be analyzed using the non-parametric Kruskal-Wallis with Dunn’s test.
Minor comments
· I suggest to add the expression data for MS4A12 confirming differentiation as a supplementary figure
· Some minor typos are present in the manuscript which would benefit from minor language editing, e.g. Line 37 “Cronh’s”, Line 114 missing “the”; line 452 ”the level” … “was”; etc.
· mtDAMPs is not defined (its clear what it means but the abbreviation should still be defined)
· Writing out all respective p-values and numeric data in the figure legends seems a bit superfluous
Author Response
Major comments
- Did the authors measure classic inflammation markers in response to the NSAID treatments? E.g. IL-6, IL-8, IL1b, TNF-a, COX-2, PGES, IL-10, CCL-20,… This seems to be missing as control for the actual effect of the treatments on the organoids. Can the authors speculate about the pathophysiology of the inflammation incurred in the genesis of IBD (autoimmune) and the effect of the NSAIDs on inflammation via mitochondrial stress?
Following the reviewer suggestion, we measured TNFα in T84 cells treated with different NSAIDs. We observed significantly higher levels in diclofenac-treated cells. Noteworthy, treatment with different NSAIDs may have a different impact on the production of pro-inflammatory cytokines. Interestingly, the effect of NSAIDs on pro-inflammatory cytokine production can be, in fact, anti-inflammatory [PMID 17083365] or paradoxically pro-inflammatory [PMID 20713883, PMID 9798931]. Therefore, we cannot determine whether the increased TNFα production in T84 cells is due to the release of mtDNA or to the direct effect of diclofenac. However, in this article, we wanted to focus on the fact that the treatment with NSAIDs induce a mitochondrial stress and further release of mtDAMPs, as measured by mtDNA levels. To deepen in this idea, we edited figure 4 to include the release of cardiolipin to the cytosol in ibuprofen-treated T84 cells, which has been recently measured in our lab. Both mtDNA and cardiolipin are widely described as DAMPs [PMID 31225514] with the ability to trigger pro-inflammatory signaling.
- Following up on this, it would be very interesting to have included in the panel of drugs also Mesalazin, which also has NSAID activity, but is used as actual treatment for IBD.
The reviewer raised an interesting point here. In fact, there is at least one previous study assessing the effects of Mesalazin on mitochondrial function (PMID 32308077). The authors observed that Mesalazin induce oxidative stress, with collapse of the mitochondrial membrane potential and increased production of mitochondrial ROS.
However, the focus of our study was not the analysis of mitochondrial function upon specific treatments for IBD. Instead, we focused on NSAIDs drugs which are consumed massively as pain-killers for the mild-to-moderate pain. Frequently, IBD patients cannot benefit from these drugs because of the risk to trigger a relapse. Our aim is to contribute to a better understanding on how these drugs trigger relapses in IBD, with the future goal of prevent these side-effects. However, as the reviewer suggests, we believe assessing the effects of drugs used for the management of IBD on mitochondrial function in intestinal epithelia would be interesting, and we will consider this idea for future projects.
- Why are Figures 3B as well as 4A and 4B shown as histograms instead of as individual datapoints? Please show all individual datapoints in all graphs (like in the other figures).
The reviewer is right, we should have been more consistent in the way we present the data. We changed these graphs, so now all graphs are scatter dot plots.
- For non-normal-distributed data (the authors mention that they use a normality test in the methods), median+-IQR should be shown instead of mean+-SD (the SD only means something if the data are normal distributed).
The reviewer is completely right about data representation. We have re-check all data:
- All groups in figure 1 are normally distributed.
- In figure 2, ATF4 expression in T84 cells treated with Paracetamol and ATF5 expression in cells treated with Paraquat did not pass the normal distribution test. We followed reviewer suggestion to show median and IQR instead of mean and SD.
- Figure 3
- In figure 4, all groups follow a normal distribution, so mean and SD are kept in the graphs.
- Up to four groups in figure 5 did not follow a normal distribution, so as correctly suggested by the reviewer, we change mean and SD for median and IQR.
- In figure 6, gene expression of CHOP and FGF21 in healthy controls show deviation from normal distribution, as well as expression of ATF4 and ATF5 in patients in remission. Therefore, following the reviewer suggestion, we plot median and IQR instead of mean and SD.
- Old supplemental figures 1, 2 and 3 have been switched to median and IQR, and colors have been applied to supplemental figure 2 and 3 to improve visualization. These figures have been re-named supplemental figure 2, 3 and 4, respectively.
- Supplemental figure 4 has been switched to median and IQR. This figure has been re-named supplemental figure 5.
- The authors should use the correct statistical tests for multiple comparisons. Specifically, the authors use multiple t-tests or Mann-Whitney U tests to compare their different treatments with the same control group. However, this does not should never be done, as it increases the likelihood of committing a type 1 error. The correct tests to use is an ANOVA with an appropriate post-hoc test (e.g. Dunnett) for normal distributed data and Kruskal-Wallis with Dunn’s for non-normal distributed data.
The reviewer is right about comparison of different problem treatments. When comparing different experimental conditions with a control group, the likelihood of type I error increases. The text in method section has been edited to describe that we applied either ANOVA + Dunnett´s test or Kruskal-Wallis + Dunn´s test, accordingly to the distribution of the data, figures and figure legends have been updated, as well as the text describing the results. However, we did a couple of exceptions:
Figure 4 and Figure 5: In fact, we are comparing Ibuprofen-treated cells with untreated cells. Paraquat is a positive control condition that we use to control the conditions of the experiment. However, the analysis of figure 5 was performed using a paired T test, which we understand was not correctly specify in the text. We used a paired test because we believe it is relevant to measure the changes in gene expression within each organoid line. And we used parametric test because most of the groups followed a normal distribution. However, as the reviewer noted, certain groups did not follow a normal distribution. We consider because we switched to plot median and IQR, it may be more consistent to used a non-parametric test. Because of that, we re-analyzed this figure using Wilcoxon test.
- In Figure 3B, the authors state they use a paired-T-test. The same issue with multiple comparisons as described above applies here, but this graph presents yet an additional problem. The Untreated control is shown as being normalized to 1. If a group does not have a standard deviation, (i.e. all values = 1 in this group), they cannot be analyzed using a parametric test. The data should instead be analyzed using the non-parametric Kruskal-Wallis with Dunn’s test.
The reviewer is again right. Although frequently found in the literature, parametric tests should be avoided when one of the groups is a repeated value (i.e. untreated cells values = 1). However, the small sample size was preventing us to find significant results even the increase of LC3B-II was clear. To support our hypothesis, we increased the sample size and perform one more immunoblot, with consistent results. The graph has been updated to show scatter dot plot and median and IQR values, and Kruskal-Wallis with Dunn´s test has been used to compared values from different groups.
Minor comments
- I suggest to add the expression data for MS4A12 confirming differentiation as a supplementary figure
Gene expression of MS4A12 in our T84 cell model has been included as supplementary figure 1.
- Some minor typos are present in the manuscript which would benefit from minor language editing, e.g. Line 37 “Cronh’s”, Line 114 missing “the”; line 452 ”the level” … “was”; etc.
The authors thank the reviewer to note these mistypes, which have been corrected.
- mtDAMPs is not defined (its clear what it means but the abbreviation should still be defined)
mtDAMPs has been spelled out upon first time of appearance.
- Writing out all respective p-values and numeric data in the figure legends seems a bit superfluous
We understand that the reading of figure legends can be cumbersome. We edited them to simplify the text and improve the readability.

Reviewer 3 Report
Comments and Suggestions for Authors
Comments and suggestions
1. Involvement of mitochondria in IBD has been previously described: swollen (less efficient) mitochondria and low levels of ATP have been associated with changes in ileum 62 permeability [6]; reduction of oxidative phosphorylation is observed in colon from IBD 63 patients [7]. If ATP first time abbreviate it.
2. Introduction should be updated with relevant study links with Mitochondrial stress links environmental triggers with pro-inflammatory signaling.
3. Figure 1. Mitochondrial membrane potential and Reactive Oxygen Species (ROS) production in T84 208 cells. (A) Data of mitochondrial membrane potential. Fluorescence (arbitrary units) was normalized 209 to percentage of untreated cells. FCCP was used as negative control in the assessment of mitochon- 210 drial membrane potential, since it is known to uncouple mitochondrial oxidative phosphorylation. 211 Mean ± standard deviation is represented. Significant values and T test results were: ibuprofen 212 (184.0%±9.5; p<1.0x10-3); diclofenac (129.9%±12.4; p<1.0x10-4); celecoxib (92.8%±10.2; p>0.05); acetyl 213 salicylic acid (103.9%±13.1; p>0.05); paracetamol (89.9%±5.3; p=5.0x10-4); all compared to untreated 214 cells (100.0%±6.5). pair t-test or unpair t-test? Please confirm.
4. It's better to abbreviate all short form in the figure legend that is used in the figure.
5. Figure 3. Assessment of different LC3B isoforms by immunoblot. This present image is not good visualization. Please use a good-resolution immunoblot image.
6. Please check all chemical reagents, manufacturer, and company name has been written correctly.
7. It should be better to avoid references in the conclusion section.
Comments on the Quality of English Language
Moderate editing of English language required
Author Response
- Involvement of mitochondria in IBD has been previously described: swollen (less efficient) mitochondria and low levels of ATPhave been associated with changes in ileum 62 permeability [6]; reduction of oxidative phosphorylation is observed in colon from IBD 63 patients [7]. If ATP first time abbreviate it.
ATP has been spelled out upon first time of appearance.
- Introduction should be updated with relevant study links with Mitochondrial stress links environmental triggers with pro-inflammatory signaling.
Two new references addressing the reviewer comments are included in the introduction section.
- Figure 1. Mitochondrial membrane potential and Reactive Oxygen Species (ROS) production in T84 208 cells. (A) Data of mitochondrial membrane potential. Fluorescence (arbitrary units) was normalized 209 to percentage of untreated cells. FCCP was used as negative control in the assessment of mitochon- 210 drial membrane potential, since it is known to uncouple mitochondrial oxidative phosphorylation. 211 Mean ± standard deviation is represented. Significant values and T test results were: ibuprofen 212 (184.0%±9.5; p<1.0x10-3); diclofenac (129.9%±12.4; p<1.0x10-4); celecoxib (92.8%±10.2; p>0.05); acetyl 213 salicylic acid (103.9%±13.1; p>0.05); paracetamol (89.9%±5.3; p=5.0x10-4); all compared to untreated 214 cells (100.0%±6.5). pair t-test or unpair t-test? Please confirm.
Since the data correspond to independent groups, the T test performed here was unpaired. We have edited the figure legend to specify this.
- It's better to abbreviate all short form in the figure legend that is used in the figure.
We failed to detect the issue the reviewer is referring to. We tried to be consistent with abbreviations in figures and figure legend. Could the reviewer be more specific, so we can detect and correct any errors in the manuscript?
- Figure 3. Assessment of different LC3B isoforms by immunoblot. This present image is not good visualization. Please use a good-resolution immunoblot image.
We understand that the reviewer does not have access to an image in high-resolution. However, we believe this may be a problem of the editorial, since we actually uploaded an image file with dimensions 3465x1407 pixels and 300ppp resolution. We are confident the image should be visible at high-resolution once published on line. However, if there is a problem with this image, we are open to facilitate a new version to the editorial team.
- Please check all chemical reagents, manufacturer, and company name has been written correctly.
Following reviewer suggestion, we checked chemical reagents and manufacturer information. Some minor changes have been applied to reagents name, but we did not find any error in manufacturer information.
- 7. It should be better to avoid references in the conclusion section.
The old reference number 33 (which was an error) has been substituted by reference number 8. Both cited references in the conclusions section are first cited in the introduction and further discussed in the discussion section. We understand the reviewer wants to make a difference between the discussion section (where literature is discussed) and conclusion section. However, when stating in the conclusion section that our study confirms previously reported data, we considered it may be useful to indicate these references again, so the reader can easily access to this previously reported data without coming back to the introduction section. We are in any case open to remove these two references from the conclusion section if the reviewer still considers it inappropriate.

Round 2
Reviewer 1 Report
Comments and Suggestions for Authors
The authors implemented the manuscript by answering my questions. In my opinion the revised manuscript is acceptable for publication.
Reviewer 2 Report
Comments and Suggestions for Authors
The authors have made very pertinent replies to the issues raised in my review. They have also supplied additional data and made all suggested changed regarding data presentation and statistical analysis. The paper and research is of very high quality and in my opinion ready for publication.
Reviewer 3 Report
Comments and Suggestions for Authors
Thanks for improving your manuscript. Hope for the best.
Thanks
Comments on the Quality of English LanguageMinor editing of English language required